# Function and Application of Flavonoids in the Breast Cancer

**DOI:** 10.3390/ijms23147732

**Published:** 2022-07-13

**Authors:** Min Yeong Park, Yoonjung Kim, Sang Eun Ha, Hun Hwan Kim, Pritam Bhangwan Bhosale, Abuyaseer Abusaliya, Se Hyo Jeong, Gon Sup Kim

**Affiliations:** 1Research Institute of Life Science, College of Veterinary Medicine, Gyeongsang National University, Gazwa, Jinju 52828, Korea; lilie17@daum.net (M.Y.P.); sangdis2@naver.com (S.E.H.); shark159753@naver.com (H.H.K.); shelake.pritam@gmail.com (P.B.B.); yaseerbiotech21@gmail.com (A.A.); tpgy123@gmail.com (S.H.J.); 2College of Nursing, Konyang University Medical Campus, 158, Gwanjeodong-ro, Seo-gu, Daejeon 35365, Korea; ykim@konyang.ac.kr; 3Biological Resources Research Group, Gyeongnam Department of Environment Toxicology and Chemistry, Korea Institute of Toxicology, 17 Jegok-gil, Jinju 52834, Korea

**Keywords:** breast cancer, apoptosis, flavonoid

## Abstract

Breast cancer is one of the top causes of death, particularly among women, and it affects many women. Cancer can also be caused by various factors, including acquiring genetic alteration. Doctors use radiation to detect and treat breast cancer. As a result, breast cancer becomes radiation-resistant, necessitating a new strategy for its treatment. The approach discovered by the researchers is a flavonoid, which is being researched to see if it might help treat radiation-resistant breast cancer more safely than an approved medicine already being used in the field. As a result, this study focuses on the role of flavonoids in breast cancer suppression, breast cancer gene anomalies, and the resulting apoptotic mechanism.

## 1. Introduction

Cancer is a condition in which some cells in the body grow out of control and spread to other parts of the body. Cancer can begin anywhere in the trillions of cells that make up the human body [1]. Various cancers such as lung, stomach, esophageal, and breast cancer exist all over the body without our knowledge. These cancers initially disappear due to the body’s immune response, but if not, the tumor grows and causes damage to the body and develops into cancer. These cancers have the following several symptoms: weight fluctuations, such as unexpected weight loss or increase; yellowing, darkening, or redness of the skin, unhealed wounds, alterations to existing moles are all signs of skin changes; as well as, coughing that will not go away or breathing problems [2].

Among cancers that are a global issue, breast cancer is the most common cancer in the world [3]. Breast cancer is a frequent condition among women in their forties and fifties. It has a 20% mortality rate and a 30% morbidity rate. Furthermore, the cancer incidence rate is increasing daily due to developing human physiological instability and current nutritional eating habits [4]. Premature menstruation and delayed menopause, for example, increase the risk of breast cancer [5]. In addition, many side effects often exist even after overcoming cancer. The main side effects of breast cancer are fatigue, loss of appetite, nausea, pain, and weight loss [6].

A cancerous cell or tissue differs significantly from a physiologically normal cell or tissue. For starters, cancer starts with a single cell. In other words, when a cell obtains several mutations and transforms into a cancer cell, it continues to divide and increase to create cancer [7]. Second, once cancer has developed, it cannot control its growth. Normal tissue stops growing when it reaches a particular size, while cancer continues to grow, often forming huge masses. Invasive growth has the following attributes: as cancer spreads, it burrows into the normal tissue around it, making surgical resection difficult [8]. Furthermore, cancer is characterized by an undifferentiated condition. Each organ’s normal tissue has unique histological traits, while cancer does not [9].

Metastasis is another trait. The spread of cancer to tissues that are not directly associated with it is known as metastasis. When metastasis is discovered, the cancer is usually categorized as terminal [10]. As a result, the existence or absence of metastasis is critical in selecting a treatment strategy.

Surgery, chemotherapy, hormone therapy, biological therapy, and radiation therapy are mainly used to treat breast cancer. Surgery is the medical practice of cutting cancerous tissue, and chemotherapy is a treatment method using particular drugs to kill cancer cells [11]. Hormone therapy is a way to block cancer cells from getting the hormones they need to grow. Biological treatments work with the body’s immune system to help fight cancer cells or control the side effects of other cancer treatments [12]. Radiation therapy uses light rays similar to X-rays to kill cancer cells [13]. There are methods of such treatment, but there are cases in which people are harmed by such treatment. Among them, side effects occur, such as loss of appetite, nausea and vomiting, fatigue, mouth pain, hair loss, weight gain, and a higher risk of infection by other diseases [14]. In addition, as a reaction to these side effects, there are cases where there is a resistance to the treatment method, so even if other treatments are used, there are cases where it is meaningless [15].

When cells grow and divide more than they should, or do not die when they should, an abnormal tissue mass arises [14]. They are called tumors. Tumors can be benign or malignant. Benign tumors grow slowly and expand and have a capsule to prevent spread to surrounding tissues. Moreover, it is well-differentiated and the cells are mature [16]. When surgically removed, recurrence is rare, and there is no metastasis. Benign tumors are almost harmless to the human body, but they become problematic when pressure is applied to significant organs or when they are closed. There is virtually no harm to the human body, and the prognosis is good. Conversely, malignant tumors proliferate and grow while infiltrating into surrounding tissues [17]. Because there is no capsule, it penetrates the surrounding tissue, making removing the tumor difficult even with surgery. They have poor differentiation and immature cells since it spreads to surrounding tissues, recurrence is common after surgery [17]. Metastasis is common because it spreads to surrounding and moving issues. If left untreated with surgery, radiation therapy, or chemotherapy, it can cause death. For malignant tumors, the prognosis depends on the time of diagnosis, degree of progression, and metastasis [18]. 

Flavonoids are a class of natural substances that have phenolic structures in diverse forms and are found in plants [19]. Flavones, flavanones, flavanols, flavonols, isoflavones, and anthocyanidins are the six subclasses of flavonoids [20]. According to a study, flavonoids have anti-inflammatory [21], antiviral [22], anti-allergic [23], antioxidant [24], and anti-tumor [25] properties. According to a study, flavonoids also inhibit tumor growth by causing death in cancer cells [26]. Therefore, it is possible to treat breast cancer more safely than a dangerous method with side effects by inducing the death of cancer cells and receiving radiation treatment.

There is much information about flavonoids and their numerous routes in breast cancer. As a result, we would like to summarize flavonoids’ anticancer properties and the links between flavonoids and breast cancer.

## 2. Flavonoid and Anti-Cancer Effect

Flavonoids are natural compounds that belong to a group of polyphenolic plant secondary metabolites found in various fruits, vegetables, and beverages [27,28]. Flavonoids are anticancer because they reduce the symptoms of cancer. Flavonoid applications include proliferation inhibition, cell cycle arrest, apoptosis, antioxidants, and anti-metastasis. The caspase-9, mitochondrial-driven apoptosis, extrinsic, caspase-8, and death receptor-driven apoptosis signaling pathways, among others, are involved in flavonoids’ anticancer activities [27].

Bcl-2 (B-cell lymphoma 2), PARP (poly ADP-ribose polymerase), FLIP (FADD (Fas-associated death domain)-like interleukin 1-converting enzyme) inhibitory protein), as well as regulatory genes such as p53, can all be affected [29]. Flavonoids such as quercetin can inhibit PKB (protein kinase B (also referred to as Akt)) [30], the protein implicated in apoptosis avoidance [31].

c-FLIP is a master regulator of anti-apoptotic mechanisms that have been found in abundance in a variety of cancer cells [32]. The overexpression of c-FLIP interferes with the caspase-8 protein, preventing caspase-8 from cleaving and terminating the apoptotic mechanism [33]. Thymoquinone reduced c-FLIP expression at the translational level but did not influence c-FLIP transcriptional regulation. However, the literature has well documented that ROS plays a role in the post-translational modification of c-FLIP via increased proteasomal activity [34,35]. Numerous researchers have recently looked into the relationship between NF-кB and Bcl-2; for example, the transcriptional expression of the NF-B target BCL-2 [36,37].

One of the fundamental mechanisms behind flavonoids’ anti-tumor action has been discovered as a cell cycle arrest. Both natural and synthetic flavonoids have been found to limit cancer cell proliferation at the G2/M phase, principally through regulating cyclin expression levels [38,39,40].

Flavonoids (Figure 1) can modulate proliferation [5,39,41,42,43,44], invasion [45,46], and inflammatory signals through a variety of pathways associated with their qualities as antioxidants [24,47], estrogen agonists, or CYP1 (Cytochrome P450) inhibitors [39]. Flavonoids have also been found to influence inflammatory signaling via systems such as NF-кB (nuclear factor B), which regulates cell proliferation and survival [21,23]. 

Many flavonoids have been shown to have anticancer properties. In human breast cancer cells, certain flavonoids (isorhamnetin, genkwanin, acacetin, silymarin, eriocitrin, icariin, kaempferol, silibinin, apigenin, and luteolin) suppress cell growth and induce apoptosis, autophagy, or cell cycle arrest [5,20,38,43,44,48,49]. Table 1 shows the anticancer properties of introduced flavonoids.

Furthermore, the anticancer action of flavonoids has been discovered to be broad (affecting most types of cancer) and cancer-specific, with levels that cause cancer cells to proapoptotic and have no effect on normal cells [29]. The flavonoids’ capacity to target common pathways in cancer cells is most likely to blame. The CYP1 enzyme family, for example, is expressed primarily in tumor cells and premalignant tissue, with no evidence of expression in normal surrounding tissues [50]. This suggests that developing medicines targeted exclusively to aberrant tissue has a lot of promise. Its specificity could be further improved because many of the chemicals employed are only bioactivated when they come into contact with these enzymes.

Flavonoids decrease cancer cell proliferation and invasiveness via modulating ROS-scavenging enzyme activities, participating in cell cycle arrest, inducing apoptosis and autophagy, and suppressing cancer cell proliferation and invasiveness [51]. 

Because of their ability to stabilize free radicals due to the presence of phenolic hydroxyl groups, flavonoids can directly scavenge ROS and chelate metal ions [52,53]. Activating antioxidant enzymes, repressing pro-oxidant enzymes, and stimulating antioxidant enzymes and phase 2 detoxification enzymes are indirect flavonoid antioxidant effects [51]. Flavonoid anticancer actions entail both antioxidant and pro-oxidant activity [54].

**Table 1 ijms-23-07732-t001:** Relationships between flavonoid and anticancer activity.

Flavonoid	Anti-Cancer Activity from the Study	Cancer	Reference
Isorhamnetin, Genkwanin, and Acacetin	Inhibit cell proliferation and induce apoptosis and autophagy	Breast cancer	[43]
Silymarin	Suppresses proliferation inducing apoptosis	Ovarian cancer	[55]
Kaempferol	Induce proliferation and induce cell cycle arrest, apoptosis, and DNA damage	Breast cancer	[38]
Eriocitrin	Inhibited cell growth and promoted death	Breast cancer	[5]
Apigenin and luteolin	Inhibit proliferation	Breast cancer	[20]
Icariin	Apoptosis and improve anti-tumor immunity	Breast cancer	[48]
Silibinin	Autophagy is triggered by ROS-dependent mitochondrial failure.	Breast cancer	[49]

Flavonoids also have an additive effect. As a result, one of the potential ways to apply flavonoids to cancer treatment has been suggested as follows: a mixture of various polyphenols administered concurrently with anticancer medicines. According to a study flavonoids do not influence other natural anticancer activities (such as phase II detoxification enzymes’ influence). A flavonoid is a good natural cancer prevention product due to its accessibility, demonstrated efficacy, and few side effects.

## 3. Breast Cancer

Cancer cells have DNA and RNA that are very similar (but not identical) to cells from the organism from whence they came. This is why they are not always noticed by the immune system, even significantly when it is compromised [56]. Cancer arises when the immune system malfunctions and/or the quantity of cells created exceeds the immune system’s ability to eradicate [57]. Under certain circumstances, such as an unfavorable environment, the rate of DNA and RNA mutations might be excessive (due to radiation, chemicals, etc.) [8].

Tumors are abnormal tissue masses that occur when cells divide and expand faster than they should or do not die when they should. Tumors can be benign (non-cancerous) or malignant (cancerous) [58]. Benign tumors can grow quite large but do not spread to surrounding tissues or other body sections [58].

In both industrialized and developing countries, breast cancer is one of the most frequent cancers among women. Around 2 million new cases of breast cancer were reported worldwide, according to the American Cancer Society [59]. In 2017, more than 250,000 new breast cancer cases were identified in the United States, and 12 percent of all women in the country will be diagnosed with the disease at some point throughout their lives [32]. Promote cell proliferation, reduce cell death, or have biological changes leading to breast cancer [20]. Breast cancer develops in the breast tissue, most usually in the inner lining of milk ducts or the lobules that supply milk to the vents. Breast cancer is 100 times more common in women than in men, yet males have a worse prognosis due to detection delays [60]. 

Breast cancer can be treated with various options, such as surgical resection with or without lymph node dissection, radiation, and chemotherapy. For breast cancer patients, adjuvant chemotherapy, radiation therapy, or hormone therapy is applied after surgery. If surgery is not possible, radiation therapy and chemotherapy are administered [61]. Chemotherapy is widely used before and after surgery for curing and reducing recurrence [62]. In this case, hormone therapy is used to lower the risk of recurrence of hormone-sensitive breast cancer [61]. At this time, in the case of systemic treatment such as chemotherapy or hormone therapy, it may cause a systemic reaction and, in the case of women of childbearing age, it may affect fertility. On the other hand, radiation therapy, a local treatment method, is applied to all patients who have had a partial mastectomy. Radiation therapy is one of the essential local and regional therapies for breast cancer treatment. Most breast cancer patients receive radiotherapy after surgical resection; however, not all patients benefit equally because some have a locoregional relapse. The leading cause (Table 2) of this relapse is radio resistance [63]. Radio-resistance is the adaptability of malignant cells or tissues to radiotherapy-induced damage and irradiation survival (IR) [64,65].

Radiotherapy usually causes severe side effects in women undergoing breast cancer treatment. Radiation therapy can cause exhaustion and a red, sunburn-like rash where radiation is being delivered. Breast tissue may appear larger or stiffer as well [63]. These negative consequences might be debilitating. Radiation-resistant breast cancer cells cannot receive radiation therapy or helpful treatment. Therefore, breast cancer should be treated using other methods, and one of those methods is to treat cancer using flavonoids [66]. In particular, using adjuvant flavonoids can lower the risk of breast cancer and systemic side effects, which can improve breast cancer survivors’ quality of life. By noting that these flavonoids have these actions in vivo, the possibility of flavonoids being treated for breast cancer is increased (Table 3 and Table 4).

**Table 2 ijms-23-07732-t002:** Relationship between breast cancer side effect case and radiation therapy.

Side Effect	Explanation	Reference
Breast changes	Radiation may cause the breasts to shrink or become denser.	[67]
Brachial plexopathy	Breast or chest wall radiation can damage the nerves that travel through the arm, wrist, and hand. Damage to the nerves can result in numbness, discomfort, or weakness in the affected area.	[68]
Sore throat	Radiation to the lymph nodes around the collarbone might produce a painful throat or make swallowing difficult. Once the treatment is over, these symptoms should go away.	[69]
Lymphedema	Lymphedema is a condition in which the arm, hand, or chest swells. Radiation can sometimes harm neighboring lymph nodes, resulting in lymph fluid accumulation.	[70]
Nausea	Radiation can produce nausea; however, this is an infrequent side effect.	[71]
Rib fracture	Radiation therapy can weaken the ribs, making them more prone to breaking or fracturing. However, with the use of new treatment regimens, this is a relatively rare occurrence.	[72]
Heart problems	The heart can be damaged if a doctor uses radiation on the left side of the chest. This is rare now that new protocols have been implemented.	[2]
Lung problems	Radiation can induce inflammation in the lungs on an infrequent occasions. Radiation pneumonitis is the medical word for this condition, which causes shortness of breath, coughing, and a low-grade fever that will go away with time.	[73]
Swelling	Swelling or inflammation of the breast or surrounding tissue is possible. Swelling should subside after a few weeks of the treatment’s completion.	[74]
A second cancer	In scarce situations, radiation exposure can raise the risk of developing second cancer.	[75]

## 4. Suppose Pathway

Flavonoids induce apoptosis as a usual way of acting on cells in radiation-resistant breast cancer. We examined simple anticancer activity from many flavonoid effects (Table 1, Table 3 and Table 4). Using such products, a non-harmful therapeutic agent can be developed to select and kill cancer cells in the field safely. An essential aspect of this treatment is that flavonoids induce apoptosis, and thus the cancer cells disappear. Except for cancer, it can be a safe therapeutic strategy for the human body. Therefore, we are going to focus on apoptosis and breast cancer.

When a cell stops growing and dividing, it enters apoptosis, a sort of cell death [55]. Within physiological conditions, in other words, apoptosis (Figure 2) is a critical mechanism that results in the eradication of unwanted cells [27]. There are those whose flavonoids can affect each of those pathways and in which layer the pathway they may exert their role. Extrinsic and intrinsic apoptosis pathways are two types of apoptosis pathways. The extrinsic route, which includes caspase-8, -10, -3, -7, and BID, is one of the apoptotic pathways (BH3-interacting domain death agonist). Apoptosis can be triggered by extracellular death receptors such as TNF-related apoptosis-inducing ligand (TRAIL), tumor necrosis factor (TNF), and Fas cell surface death receptor (FAS), as well as intracellular stimuli such as hunger, irreversible genetic damage, osmotic stress, and hypoxia [91]. The intrinsic pathway, also called the mitochondrial apoptosis pathway, comprises many stimuli that act on different cell targets. Caspase-8 and caspase-10 can BID shortened BID in the extrinsic route. The tBID (to be decided or to be determined) activates BAK (BCL2-antagonist/killer 1) and BAX (Bcl-2-associated X protein) [55]. Inherent lethal stimuli can activate BAX and BAK via stimulating BH3-only proteins. Activated BAK and BAX can form MPT (mitochondrial permeability transition) pores on mitochondrial outer walls, allowing cytochrome c to leak into the cytoplasm due to MOMP induction and act as a signaling molecule in the cytoplasm, facilitating the formation of apoptosomes (proapoptotic proteins such as Smac/Diablo, HrtA2/Om, and cytochrome c). When an apoptosome forms, caspase-9 is the first to be activated, followed by caspase-3 and caspase-7. Activating caspase-3 and caspase-7 causes apoptosis, killing cellular components [55,92].

The extrinsic pathway contains genes from the superfamily and includes interactions mediated by transmembrane death receptors, TNF receptors, and the extrinsic pathway. The creation of a “death-included signaling complex” (DISC) comprised of TNF receptor 1 (TNFR1), Fas-Fas-L, death receptor 3 (DR3), death receptor 4 (DR4), and tumor necrosis factor superfamily 10 (known as TRAIL/Apo2L) is triggered by binding between death receptors and ligands [92]. Various death receptors and ligands are found in the extrinsic route. As examples, we will look at Fas-L and TNF receptors. When FAS is combined with Fas-L, it forms a trimer. As a result, the FAS death effector domain becomes visible, which can be linked to the adaptor protein FAS-associated death domain (FADD) [93]. Procaspase-8 and procaspase-10 can bind to this domain, cleaving them to activate caspase-8 and caspase-10, which subsequently activate caspase-3 and caspase-7 as effector caspases, breaking the target protein and producing apoptosis [55].

Flavonoids have been utilized to treat cancer by inducing apoptosis in several trials. In Hep3B cells, scutellarein can trigger the Fas-mediated extrinsic apoptotic pathway [94]. Apigenin increases cell death in MDA MB-231 and MCF-7 human breast cancer cells, causing significant toxicity and, most critically, apoptosis [95]. Apoptosis is also induced by GL-V9 in human breast cancer cell lines [96]. By reducing miR-27a expression, isoliquiritigenin, a flavonoid, can also block melanoma cells from growing and migrating [55,91].

Normal breast development is controlled by a balance of cell proliferation and apoptosis, and there is evidence that tumor growth is caused by excessive proliferation and decreased apoptosis [97]. The balance of proliferation and apoptosis is critical in deciding whether a tumor will grow or reduce in response to chemotherapy, radiation, or hormone therapy [63,98]. All of these things work by causing apoptosis in some way. Investigating apoptosis and its control and treatment makes it feasible to characterize the biology of specific tumors at the molecular and biochemical levels and use this information to assist patients [99]. It can be said that this is an effect that can be obtained by inducing apoptosis by flavonoids to such an extent that the patient does not feel uncomfortable even after treatment in the direction of fewer sequelae than helping the patient. Certain drawbacks to these trials could make it difficult to say definitively that they have an anticancer impact.

Genetic mutations accumulate in epithelial tissue during carcinogenesis, and cellular functions are lost. The phenotypic characteristics of cells alter as they progress from average to malignant lesions, superficial tumors, and finally, invasive illness [100]. During the premalignant stage, apoptosis, proliferation, and cell cycle regulatory markers differ significantly. Apoptosis is enhanced in both ductal carcinomas in situ and invasive breast cancers [101]. Invasive breast cancer apoptosis appears lower than proliferation in normal breast epithelium [27,99].

The cell death pathway’s fundamental mechanism could be simplified to just a few essential proteins that a conserved across species. In humans, these regulators have been discovered with many homologs expressed in various tissues [99,102]. In the context of breast cancer, the Bcl-2 family of apoptosis inhibitors and promoters and the p53 tumor suppressor gene have been intensively researched [103]. The image demonstrates the relationship between these oncogenes and apoptotic pathway proteins (Figure 3).

Proteins produced by the Bcl-2 gene family can either promote or inhibit apoptosis. Pro-apoptotic proteins include Bax, Bak, Bad, and Bcl-xs, whereas anti-apoptotic proteins include Bcl-2 and Bcl-xL (B-cell lymphoma-extra-large). Bcl-2 is connected to the expression of estrogen and progesterone receptors, both favorable prognostic markers in breast cancer, and is expressed in about 75% of primary breast cancer [104] malignancies. Patients with Bcl-2 positive tumors have a better prognosis than those with Bcl-2 negative cancers, suggesting an unexpected link between an apoptotic inhibitor and a favorable result. BAG-1 (BAG family molecular chaperone regulator 1), a multifunctional protein that controls apoptosis, has recently been linked to better survival in women with early-stage breast cancer [104]. BAG 1 interacts with other members of the Bcl-2 family, as well as heat shock proteins and estrogen receptors, in what appears to be a contradictory finding in a protein.

P53 (Figure 4) is a protein that regulates various biological processes and cell cycles, including apoptosis, biological processes, and DNA repair [105]. The most prevalent mutational event in cancer is mutations in this gene. Several studies have connected gene mutations or increased p53 protein synthesis (an indirect indicator of conversion because it typically leads to protein stabilization) to a poor prognosis in breast cancer [106,107]. Chemotherapy, tamoxifen, and radiotherapy have been shown to cause apoptosis in cells via p53-dependent and p53-independent routes [108,109].

## 5. Conclusions

Flavonoids are a broad group of physiologically active water-soluble plant compounds (such as anthocyanins and flavones) that are prevalent in fruits, vegetables, and herbs and come in various colors ranging from yellow to red to blue [110]. Flavonoids have many anticancer functions, such as proliferation, invasion, and inflammatory signals through various pathways associated with their qualities as antioxidants [24]. These flavonoids can treat cancer cells by targeting their anticancer effects.

Breast cancer is one of the most common malignancies in women, both in terms of incidence and death, and it is also one of the deadliest diseases on the planet [111]. Radiation is often used when examining or treating breast cancer. The radiation makes breast cancer cells resistant to radiation. In this case, a treatment method other than radiation should be chosen, and we think that flavonoids in combination with other treatments is suitable among those methods. Not only is it safe, but because it induces the cell’s function, flavonoids can be a safe method unlike regular anticancer drugs that attack other parts of the cell.

Apoptosis (Figure 5) in breast cancer has been studied using the Bcl-2 family of tumor suppressors and the p53 tumor suppressors. Bcl-2 proteins are members of the Bcl-2 family of proteins. Bcl-2 and Bcl-xL are antiapoptotic, whereas Bax, Bak, Bad, and Bcl-xs are proapoptotic [55]. BAG-1, a multifunctional apoptosis-controlling protein, has lately been related to an improved prognosis in patients with early-stage breast cancer [104]. Apoptosis, the cell cycle, and DNA repair are all regulated by the protein p53. It causes apoptosis in both p53-dependent and non-dependent cells [112,113].

Women who have had breast cancer up to this point have experienced side effects even after being cured by treatment (Table 2). To avoid these side effects, if flavonoids were commercially available as a drug without side effects in an environmentally friendly way and used for various cancer patients, many cancer patients would be able to recover and take care of their lives fully. Flavonoids are necessary for you to be as comfortable as possible while avoiding any adverse effects. Moreover, to affirm the antitumor impact, we believe it is vital to identify limitations that may occur in this research.

The most frequent malignancy among women is breast cancer. It may cause discomfort if the breast develops resistance to radiation therapy and the tumor is treated by sectioning the breast. Therefore, this review paper proposes a method for treating cancers, including those that have acquired radiation resistance, in combination with flavonoids without causing inconvenience to patients with these cancers, especially breast cancer, as in this review paper.

## Figures and Tables

**Figure 1 ijms-23-07732-f001:**
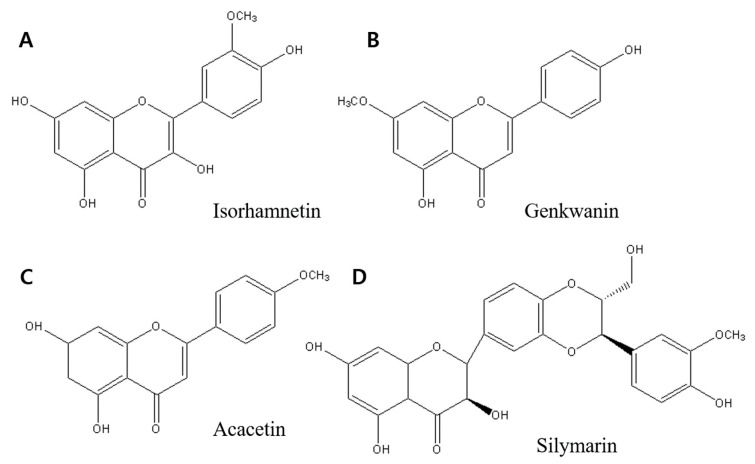
Structure of flavonoid. They were drawn by ChemDraw Pro 8.0.

**Figure 2 ijms-23-07732-f002:**
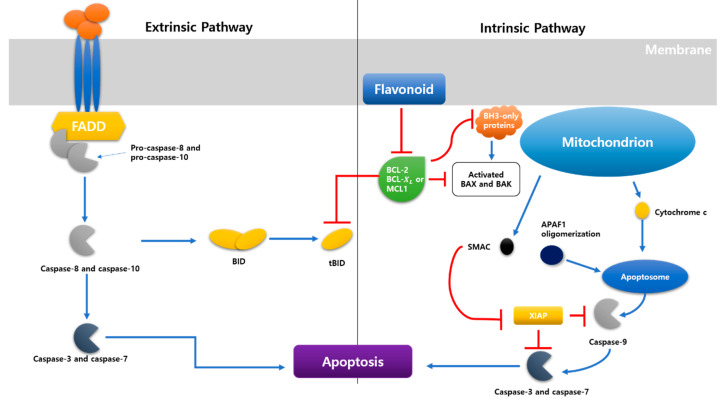
There are two pathways of apoptosis. There are extrinsic and intrinsic pathways, and both induce apoptosis.

**Figure 3 ijms-23-07732-f003:**
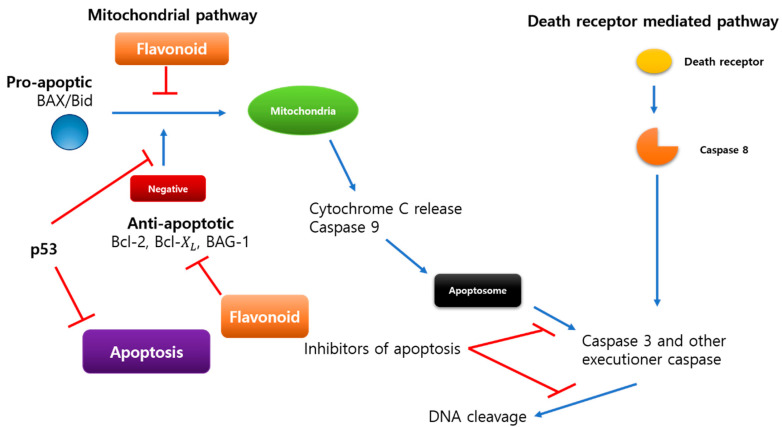
Between oncogenes and proteins involved in apoptosis. It was discovered that there is a link between chemicals that inhibit apoptosis and the consequences that cause it.

**Figure 4 ijms-23-07732-f004:**
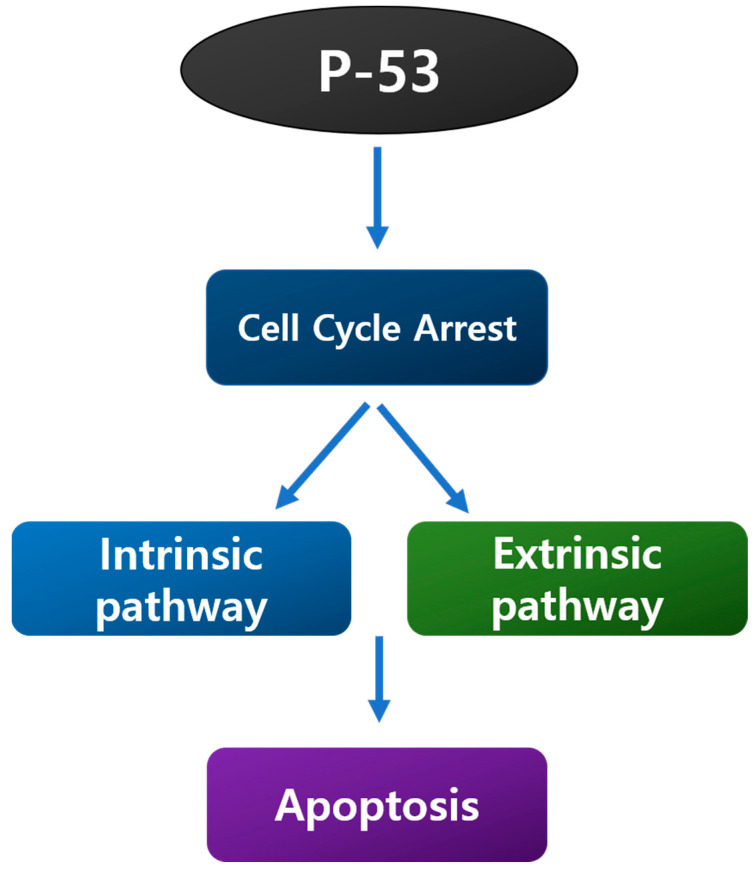
Biological processes involving P53.

**Figure 5 ijms-23-07732-f005:**
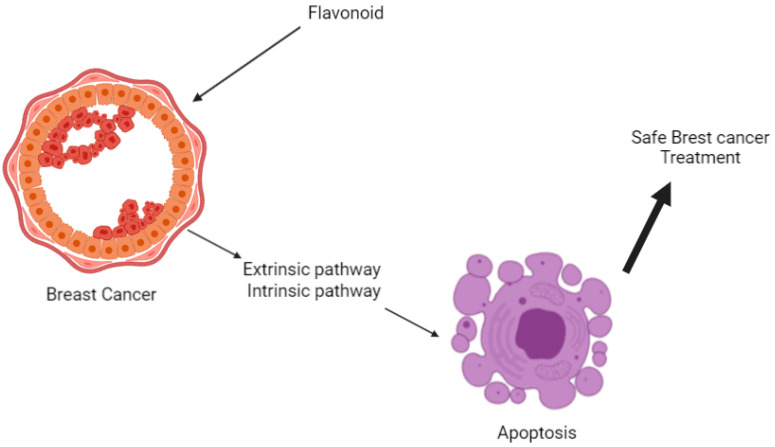
Flavonoid-induced apoptosis of breast cancer cells.

**Table 3 ijms-23-07732-t003:** The principal effect of flavonoid in vivo.

Flavonoid	Function In Vivo in Breast Cancer	Reference
Kaempferol	Osteoprotective effect	[76]
Apigenin	Inhibit tumorigenesis	[77]
Icariin	Prevent glucocorticoid-induced osteocyte apoptosis	[78]
Luteolin	Anticarcinogenic effect Antioxidant	[79,80]
Acacetin	Exhibit anticancer activity Inhibit angiogenesis	[81,82]
Scutellarein	Inhibition of cell proliferation and Metastasis	[83]

**Table 4 ijms-23-07732-t004:** Relationship between flavonoid and plant.

Flavonoid	Plant	Reference
Scutellarein	*Scutellaria baicalensis*	[83]
Prunetinoside	*Prunus yedoensis*	[84]
Naringin	*Citrus grandis*	[85]
Apigetrin	*Teucrium gnaphalodes*	[86]
Anthocyanins	*Vitis cognitive Pulliat*	[87]
Luteolin	*Cichorium endivia*	[88]
Kaempferol	*Vitis vinifera* L.	[89]
Silibinin	*Cirsium japonicum var. ussuriense* (*Regel*) *Kitam.*	[90]

## Data Availability

Not applicable.

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
