# Peer review of "Function and Application of Flavonoids in the Breast Cancer"

_ijms, 2022, doi:10.3390/ijms23147732_

Round 1

Reviewer 1 Report

In the manuscript “Function and Application of Flavonoid in the Breast Cancer” by Park et al., the authors make a revision on the properties of flavonoids that confer it anticancer properties, particularly focused in breast cancer.

In my opinion, I consider that the authors have made an effort in the revision, but a very substantial improvement in the manuscript is needed before it can be considered for publication. 

Below, I mention a few points that substantiate my opinion:

1. Massive English editing is required.

2. There is a clear lack of bibliographic references for many of the statements throughout the manuscript.

3. An overly simplistic (and even too basic) description is given of what tumor establishment and development/progression is, and in particular breast cancer. Having a relevant role in the theme of the article, it is necessary to have something more substantial and based on important publications. In addition, several theories on tumor establishment must be discussed in detail. 

4.1. There are certain paragraphs where the authors mention that there are several interactions that can promote antitumor effects, but in a too fast and simplistic way. Just as one of many examples:

Bcl-2 (B-cell lymphoma 2), PARP (poly ADP-ribose polymerase), FLIP (FADD (Fas-associated death domain)-like interleukin 1-converting enzyme) inhibitory protein), and FLIP (FADD (Fas-associated death domain)-like interleukin 1-converting enzyme) inhibitory protein), as well as regulatory genes like p53, can all be affected [19]. Flavonoids  like quercetin can inhibit PKB (protein kinase B (also referred to as Akt)) [20], the protein implicated in apoptosis avoidance [21].”

4.2. Why are these proteins or pathways important? What are their role in cancer? How do these proteins are regulated by flavonoids and which ones? Is it a positive or negative regulation? Is it mediated by inhibition of activity or by reduction of expression? Are there some particular tumor types that are (or might be) susceptible to those antitumor effects based on key signalling pathways?

4.3. Table 1 might be substantially improved based on the comments from bullets 4.1 and 4.2. 

4.4. The same for table 3. What description/observation from the bibliograpgy substantiate those described functions in vivo?

5. The description of the effects that have antitumor potential are described in a very generic way, without specifying in which types of cancer, if it was tested in vitro or in vivo, and exactly what mechanisms are behind these effects. In addition, I consider that it is necessary to mention limitations that may exist in these studies and that may be limiting to categorically affirm the antitumor effect.

6. Table 2 must also be improved. Please add more columns about the patient characteristics where those side effects were observed. And also, which dose or type of radiation was used.

7. In the “Suppose Pathway” section, authors describe the instrinsic and extrinsic apoptotic pathways as probable pathways by which flavonoids may exert their antitumoral effect. Please add description on whose flavonoids can affect each of those pathways and in which layer od the pathway they may exert their role.

7.2. In the same way, it can be also used to improve figure 2. As it is, the figure just represents the apopototic pathway, mostly seen in many books and reviews. The same for figure 3: It can be improved by indication what flavonoids affects that pathways!

Despite the comments, I consider that the subject of the review is important and relevant, it just needs to be well developed. For this reason, I think authors should use the comments as a positive way to help them improve their work so that it can be published and it can be an important article in this area and a reference for other researchers.

Author Response

In the manuscript “Function and Application of Flavonoid in the Breast Cancer” by Park et al., the authors make a revision on the properties of flavonoids that confer it anticancer properties, particularly focused in breast cancer.

In my opinion, I consider that the authors have made an effort in the revision, but a very substantial improvement in the manuscript is needed before it can be considered for publication. 

Below, I mention a few points that substantiate my opinion:

  1. Massive English editing is required.

Answer: Thank you for your suggestion. I let read to lab member and had correction

  1. There is a clear lack of bibliographic references for many of the statements throughout the manuscript.

Answer: Thank you for your suggestion. We added a reference paper.

  1. An overly simplistic (and even too basic) description is given of what tumor establishment and development/progression is, and in particular breast cancer. Having a relevant role in the theme of the article, it is necessary to have something more substantial and based on important publications. In addition, several theories on tumor establishment must be discussed in detail. 

Answer: Thank you for your suggestion. We added more information in manuscript

4.1. There are certain paragraphs where the authors mention that there are several interactions that can promote antitumor effects, but in a too fast and simplistic way. Just as one of many examples:

“Bcl-2 (B-cell lymphoma 2), PARP (poly ADP-ribose polymerase), FLIP (FADD (Fas-associated death domain)-like interleukin 1-converting enzyme) inhibitory protein), and FLIP (FADD (Fas-associated death domain)-like interleukin 1-converting enzyme) inhibitory protein), as well as regulatory genes like p53, can all be affected [19]. Flavonoids like quercetin can inhibit PKB (protein kinase B (also referred to as Akt)) [20], the protein implicated in apoptosis avoidance [21].”

Answer: Thank you for your suggestion. We added more explanation.

4.2. Why are these proteins or pathways important? What are their role in cancer? How do these proteins are regulated by flavonoids and which ones? Is it a positive or negative regulation? Is it mediated by inhibition of activity or by reduction of expression? Are there some particular tumor types that are (or might be) susceptible to those antitumor effects based on key signalling pathways?

Answer: Thank you for your suggestion. The reason why these protein pathways are important is that cancer can safely treat cancer by causing apoptosis that self-destructs without external factors. Each flavonoid is different, but it controls cancer by blocking MAPK or other upstream factors in the pathway, or by blocking the receptor itself, inducing substances related to mitochondria. This is generally a negative regulation, and various substances induce apoptosis due to inhibition of activity and reduction of expression. There is no set specific tumor type. Among cancer cells, we think that it is impossible to specify because the effects on various cancer cell lines or cells, such as breast cancer, stomach cancer, and liver cancer, cannot be the same.

4.3. Table 1 might be substantially improved based on the suggestions from bullets 4.1 and 4.2. 

Answer: Thank you for your suggestion. We tried that to improve

4.4. The same for table 3. What description/observation from the bibliograpgy substantiate those described functions in vivo?

Answer: Thank you for your suggestions. We changed Function in Vivo to Function in Vivo in breast cancer. By noting that these flavonoids have these actions in Vivo, the possibility of flavonoids being treated for breast cancer is increased.

  1. The description of the effects that have antitumor potential are described in a very generic way, without specifying in which types of cancer, if it was tested in vitro or in vivo, and exactly what mechanisms are behind these effects. In addition, I consider that it is necessary to mention limitations that may exist in these studies and that may be limiting to categorically affirm the antitumor effect.

Answer: Thank you for your suggestion. We added the sentence more

  1. Table 2 must also be improved. Please add more columns about the patient characteristics where those side effects were observed. And also, which dose or type of radiation was used.

Answer: Thank you for your suggestion. We think it is impossible to write down each patient's case because the side effects do not increase in one case and several side effects appear at once.

  1. In the “Suppose Pathway” section, authors describe the instrinsic and extrinsic apoptotic pathways as probable pathways by which flavonoids may exert their antitumoral effect. Please add description on whose flavonoids can affect each of those pathways and in which layer the pathway they may exert their role.

Answer: Thank you for your suggestion. We add that in manuscript

7.2. In the same way, it can be also used to improve figure 2. As it is, the figure just represents the apopototic pathway, mostly seen in many books and reviews. The same for figure 3: It can be improved by indication what flavzonoids affects that pathways!

Answer: Thank you for your suggestion. We add flavonoid in figures

Reviewer 2 Report

The review paper by Kim and coworkers describes flavonoids as an adjuvant for the breast cancer therapy. This manuscript is recommended for publication after addressing the below minor issues.

1. Texts in tables include different types of descriptions (words and sentences) and thus should be made uniform.

2. There are a lot of explanations about the breast cancer as compared to the effect of flavonoids on it. I think that there should be more emphasis on the roles of flavonoids.

3. The conclusion section is too long.

4. The language in the paper is often awkwardly written and needs to be revised to produce a clearly written paper. The work needs expert in English to review and correct it.

Author Response

The review paper by Kim and coworkers describes flavonoids as an adjuvant for the breast cancer therapy. This manuscript is recommended for publication after addressing the below minor issues.

  1. Texts in tables include different types of descriptions (words and sentences) and thus should be made uniform.

Answer: Thank you for your suggestion. We think some table needs sentences and some other table needs words

  1. There are a lot of explanations about the breast cancer as compared to the effect of flavonoids on it. I think that there should be more emphasis on the roles of flavonoids.

Answer: Thank you for your suggestions. We wrote more about role of flavonoids

  1. The conclusion section is too long.

Answer: Thank you for your suggestion. We remove some sentence in conclusion section.

  1. The language in the paper is often awkwardly written and needs to be revised to produce a clearly written paper. The work needs expert in English to review and correct it.
    Answer: Thank you for your suggestion. I asked correction to lab member in English

Reviewer 3 Report

The manuscript entitled "Function and Application of Flavonoid in the Breast Cancer" by Park et al.

Comments:

- First of all, the manuscript is suffered from serious English language issues. In my opinion, the manuscript needs major language editing. There are lots of sentences with English issues, starting right from the article title.

- Actually similar topics are overwhelming in the literature, there is lack of novelty.

- The manuscript is written in a too general scope and the content is not specific.

- It is unclear why flavonoid stands out from all the other natural compounds, the authors should elaborate more on the advantages of  and potential drawbacks of flavonoids over other natural compounds.

- Many of the figures are too general and simple which are unrelated to the subject matter, these should be removed or redesigned.

- The chemical names should be marked directly under each chemical structure in Fig. 1.

- The information in the tables are messy and not focused. I suggest the authors to redesign and enrich the content.

- Typos can be found all over the manuscript.

Author Response

The manuscript entitled "Function and Application of Flavonoid in the Breast Cancer" by Park et al.

Comments:

- First of all, the manuscript is suffered from serious English language issues. In my opinion, the manuscript needs major language editing. There are lots of sentences with English issues, starting right from the article title.

Answer: Thank you for suggestion. I sent to lab member who use English and corrected.

- Actually, similar topics are overwhelming in the literature, there is lack of novelty.

Answer: Thank you for suggestion.

- The manuscript is written in a too general scope and the content is not specific.

Answer: Thank you for suggestion. We wrote more sentence to be specific.

- It is unclear why flavonoid stands out from all the other natural compounds, the authors should elaborate more on the advantages of and potential drawbacks of flavonoids over other natural compounds.

Answer: Thank you for suggestion. We added sentence advantages sentence

- Many of the figures are too general and simple which are unrelated to the subject matter, these should be removed or redesigned.

Answer: Thank you for your suggestion. We modified figures and tables.

- The chemical names should be marked directly under each chemical structure in Fig. 1.

Answer: Thank you for your suggestion. We marked as your suggestion.

- The information in the tables are messy and not focused. I suggest the authors to redesign and enrich the content.

Answer: Thank you for your suggestion. We modified tables

- Typos can be found all over the manuscript.

Answer: Thank you for your suggestion. We checked it.
